# Temperature-Dependent Structural Variability of Prion Protein Amyloid Fibrils

**DOI:** 10.3390/ijms22105075

**Published:** 2021-05-11

**Authors:** Mantas Ziaunys, Andrius Sakalauskas, Kamile Mikalauskaite, Ruta Snieckute, Vytautas Smirnovas

**Affiliations:** Life Sciences Center, Institute of Biotechnology, Vilnius University, LT-10257 Vilnius, Lithuania; mantas.ziaunys@gmc.vu.lt (M.Z.); andrius.sakalauskas@gmc.vu.lt (A.S.); kamile.mikalauskaite@gmc.vu.lt (K.M.); ruta.snieckute@chgf.stud.vu.lt (R.S.)

**Keywords:** amyloids, prion proteins, protein aggregation, fibril structure

## Abstract

Prion protein aggregation into amyloid fibrils is associated with the onset and progression of prion diseases—a group of neurodegenerative amyloidoses. The process of such aggregate formation is still not fully understood, especially regarding their polymorphism, an event where the same type of protein forms multiple, conformationally and morphologically distinct structures. Considering that such structural variations can greatly complicate the search for potential antiamyloid compounds, either by having specific propagation properties or stability, it is important to better understand this aggregation event. We have recently reported the ability of prion protein fibrils to obtain at least two distinct conformations under identical conditions, which raised the question if this occurrence is tied to only certain environmental conditions. In this work, we examined a large sample size of prion protein aggregation reactions under a range of temperatures and analyzed the resulting fibril dye-binding, secondary structure and morphological properties. We show that all temperature conditions lead to the formation of more than one fibril type and that this variability may depend on the state of the initial prion protein molecules.

## 1. Introduction

Amyloidogenic protein aggregation into insoluble, beta-sheet rich fibrils is linked with the onset of several neurodegenerative disorders, such as Alzheimer’s, Parkinson’s or prion diseases [1,2]. The way these structures form and propagate is still not fully understood, as evidence for new aggregation mechanisms or fibril structural features [3,4,5] keeps appearing on a regular basis. This lack of crucial information is likely one of the factors that has led to countless failed clinical trials [6] and to only a handful of effective, disease-modifying drugs [7]. Considering that amyloid diseases affect millions of people worldwide and the number is expected to continuously increase [8,9], it is of vital importance to gain a better understanding of the intricacies of amyloid aggregation.

One of the more interesting aspects of amyloid aggregation is the ability of a single type of protein to associate into multiple conformationally-distinct fibrils [10]. Such a phenomenon was observed with prion proteins, both in vivo and in vitro [11,12,13,14] and later–with other amyloidogenic proteins, such as amyloid beta [15], alpha-synuclein [16,17] and insulin [18,19]. These distinct conformation fibrils possess specific replication rates [20,21], morphologies [22], secondary structures [16,23] and stabilities [24]. In addition, some of these parameters are either codependent or change even after fibrils have been formed. It has been shown using computational methods that there is a correlation between the rate of self-replication and fibril symmetry, and stability [25,26]. It was also observed that oligomeric or protofibrillar intermediate aggregates gain mechanical stability when converting to fully formed fibrils [27].

Such a variation in their properties is likely one of the main aspects why potential drug candidates appear effective under a certain set of conditions, while being completely inefficient in others [28]. It has been observed in multiple studies that the conformation of resulting fibrils depends highly on the environmental conditions, under which the protein is aggregated [29,30]. These conditions include temperature [31], agitation [32], pH [18], ionic strength [33], denaturant concentration [14] and protein concentration [34]. In our recent study, we have also shown that prion proteins are capable of forming two distinct conformation fibrils under identical conditions [23]. This hints at a possibility that primary nucleation, a process during which the initial aggregation centers from, may randomly generate distinct nuclei, only some of which can further propagate under the given environmental conditions.

Determining conformational differences between amyloid fibrils is typically done by evaluating their morphological features [35], such as aggregate height, width, length and periodicity patterns, and by acquiring information about their secondary structure [36]. The methods used for this, namely atomic force microscopy (AFM) [35] and Fourier-transform infrared spectroscopy (FTIR) [37], are difficult to apply in situations, where a large number of samples have to be differentiated. In recent years, both high-throughput screening platforms [38] and computer-aided molecule design systems [39] have advanced enough to identify neurodegenerative-disease related protein–ligand interactions and possible mechanisms of fibril formation. However, since protein aggregation is still not fully understood, such methods are not ideal for determining the highly complex structure of fully formed aggregates. As an alternative, the selective and conformation-specific binding of a fluorescent probe, thioflavin-T (ThT) [23,40,41], may be used to detect different types of aggregates. Such a method was applied in our aforementioned study with prion proteins and it was efficient at identifying different types of insulin fibrils as well [40]. Applying this type of initial screening would allow to sort out fibril samples with distinct conformations in a much larger scale assay.

Prion protein aggregation experiments are conducted under a variety of conditions in vitro, ranging from ambient temperature [42] to well-above physiological temperatures [43]. In order to determine if this environmental factor has an effect on prion protein fibril variability, we tracked the aggregation of a mouse prion protein under a range of different temperatures, using a large identical sample size to evaluate their seemingly random, conformational variations. To achieve complete fibrillization in a reasonable time frame, vigorous, fragmentation-inducing agitation was also used for every temperature condition. A ThT-assay was then employed as an initial means of sorting the distinct fibril types, which were then examined using FTIR and AFM.

## 2. Results

A large sample size of mouse prion protein fragment (89–230) was aggregated under a range of temperatures (from 25 to 65 °C) with all other conditions, such as buffer solution, protein concentration and volume remaining identical. Plotting the lag time (t_lag_) dependence on aggregation temperature (Figure 1A) revealed a discontinuity in the linear trend at 45 °C. This is caused by the protein transitioning from being in mostly folded states (at temperatures below 45 °C) and mostly unfolded (above 45 °C) [43]. This is further supported by visualizing the data in an Arrhenius plot (Figure 1B), where the activation of nuclei formation was (78.7 ± 7.0) kJ/mol at low and (30.0 ± 5.0) kJ/mol at high temperatures. This transition temperature is in line with previously reported data, [43] where a shift in activation energy was also observed.

When examining the fluorescence intensity of fibril-bound ThT molecules (when all samples were cooled down to 25 °C; Figure 1C), two interesting aspects are observed. First, there was a significantly larger amount of high fluorescence intensity samples (300–600 a.u.) when the protein was aggregated at temperatures below 45 °C, when compared to higher temperatures, where only a single high intensity sample was seen. Secondly, there was also a discontinuity of fluorescence intensity values occurring at 45 °C. The one-way analysis of variance (ANOVA) statistical analysis of the data (*n* = 90 for each case) using a Bonferroni means comparison (significance level–0.01) revealed that there was a difference between the 35–45 °C sample and 50–60 °C sample fluorescence intensity distributions, where 35–45 °C fibril-bound ThT possesses a considerably larger average signal intensity. Interestingly, there was no significant difference between the lowest and highest temperature sample sets (significance level–0.01), apart from the 25 °C samples having a greater standard deviation due to the existence of a few high fluorescence intensity samples. Both of these aspects could be explained by a larger fibril concentration or the formation of superstructural fibril assemblies, however, that is not the case. Each sample contained an identical concentration of prion protein and their aggregation kinetic curves reached a plateau, indicating a finished fibrillization process. Afterwards, each sample was diluted with the reaction buffer (containing 100 µM ThT) and sonicated, which removes the possibility of large aggregate structures entrapping the dye molecules [44] or certain samples having hydroxylated ThT [45]. This leaves the option of different samples containing distinct fibrils, which have conformation-specific dye-binding.

In order to separate and identify distinct fibril samples, an excitation–emission matrix (EEM) of each sample‘s fibril-bound ThT was scanned. The fluorescence emission intensity “centers of mass” were then compared for all temperature conditions (Figure 2). We can see that in all cases, there was a cluster of these positions with varying size and dispersion. In the case of samples from 25 to 45 °C (Figure 2A–E), the clusters were more compact than their higher temperature counterparts (Figure 2F–I), with the highest dispersion observed in the 50 °C sample set (Figure 2F).

An important thing to notice is that each sample set contained several points, which did not belong to the main cluster. These outliers had quite extreme excitation/emission wavelength shifts when compared to their respective cluster, and very high or low fluorescence intensity values. In some of the sets, there were samples with 10 nm differences in their excitation wavelength (Figure 2D,G). Since there were multiple outliers in each case, and different fluorescence intensity samples within the main cluster, four most distinct EEM outliers (Figure 2 (red), Appendix B Table A1) and four main cluster samples from lowest to highest fluorescence (Figure 2 (green), Appendix B Table A1) were chosen for further structural examination. Each sample was then replicated to both increase the mass of available aggregates, and to test their self-replication propensity. In general, no significant variations in the rate of replication were observed (Appendix B, Figure A1A–I), however, most samples retained their conformation-specific fluorescence emission intensities after aggregation, especially visible in the case of lower temperature samples (Appendix B, Figure A1).

The replicated outlier (Figure 2 (red), Appendix B, Table A1) and cluster (Figure 2 (green), Appendix B, Table A1) samples were then examined using Fourier-transform infrared spectroscopy. In the case of the different fluorescence intensity cluster samples, three distinct secondary structures were observed. The second derivative FTIR spectra of aggregates prepared at low temperature conditions (Figure 3A,B, blue lines) display two minima at 1628 cm^−1^ and 1615 cm^−1^. The band at 1615 cm^−1^ is associated with stronger hydrogen bonding within fibril beta-sheets, while the band at 1628 cm^−1^ is associated with the presence of weaker hydrogen bonds [46]. The ratio between both of these positions is not identical for all samples, with the lower fluorescence intensity sample having a less expressed minima at 1628 cm^−1^, suggesting that stronger hydrogen bonds within the fibril result in less bound-ThT or a lower fluorescence quantum yield.

The 30–65 °C sample set clusters (Figure 3B–I) contain a second type of fibrils, with the main second derivative FTIR spectrum minima located at 1626 cm^−1^, which suggests a single dominant type of hydrogen bonding within the fibrils. This aggregate conformation was the most abundant, as it was observed in eight out of the nine cluster sample sets. The third fibril conformation was only observed at higher temperatures, namely 55 °C (Figure 3G) and 60 °C (Figure 3H). Similar to the low temperature fibril conformation, two minima associated with beta-sheet hydrogen bonding are observed, however, unlike in the case of fibril prepared at low temperature, the second minima were at 1618 cm^−1^, rather than 1615 cm^−1^, which indicates a weaker mode of hydrogen bonding.

When the outlier fibril samples (Figure 2 (red), Appendix B, Table A1) were replicated and centrifuged, certain sample aggregate pellets were semitransparent and gel-like. Coincidentally, their FTIR spectra displayed a noticeable band associated with guanidine hydrochloride, despite multiple centrifugation and resuspension steps. Such abnormalities were only observed in the case of fibrils prepared under 25 °C and 30 °C temperatures. One likely explanation for this event is that this type of fibril conformation is capable of trapping some of the denaturant present in solution during aggregation, either in fibril cavities or by associating into this gel-like structure. While this, in itself, indicates a different type of aggregate, in order to acquire comparable FTIR spectra, the samples had to be washed three additional times (the second derivative FTIR spectra of these gel-like samples are marked with a * symbol in Figure 4). Certain low-temperature samples also contain a very small band at 1604 cm^−1^, which means that they may contain a minor concentration of GuHCl as well.

The outlier sample second derivative FTIR spectra (Figure 4) were significantly more diverse than in the case of the main cluster samples. While most spectra shared similar minima positions to the three cluster fibril conformation spectra (Figure 4A–I), color-coded accordingly), the band intensity ratios experienced a considerably larger variation, which suggests the presence of multiple aggregates with distinct secondary structures. This is best seen in the case of 45 °C outliers (Figure 4E), where all spectra minima had similar positions, yet the 1628 cm^−1^ and 1618 cm^−1^ minima ratios were different. There was also a lot less temperature-dependent conformation distribution, as the structure associated with fibrils prepared under high temperatures was observed in the 35 °C outlier sample (Figure 4C), as was the structure associated with fibrils prepared under low temperatures.

Interestingly, there were four outliers that did not resemble any of the three main fibril conformations. The outlier generated at 25 °C (Figure 4A, black line) had a more significant minimum at 1664 cm^−1^, which is associated with the presence of turn/loop motifs in the fibril structure [46]. It also had two minima at 1622 cm^−1^ and 1615 cm^−1^, which means that, along with the strong type of hydrogen bonding, there was also a type of bonding that was dissimilar to all other fibrils. The outlier generated at 40 °C (Figure 4D, black line) shares some similarities to the high-temperature conformation, however, it has more minima associated with turn/loop motifs. The outlier generated at 55 °C (Figure 4G, black line) had a minimum at 1664 cm^−1^, similarly to the 25 °C outlier, however, it has the most intense band at 1618 cm^−1^ with a smaller minimum at 1630 cm^−1^, which means that the fibril structure had both a large amount of strong hydrogen bonds between beta-sheets, and some weak bonding. The 60 °C outlier (Figure 4H) had similarities to the 55 °C outlier, with both having significant minima at 1664 cm^−1^, however, the 1630 cm^−1^ position is more of a shoulder, rather than a minimum, which indicates slightly less weak hydrogen bonds (these two samples were regarded as similar outliers). All four samples that had a gel-like appearance (marked with * in Figure 4) share minima positions with the high temperature conformation, however, their FTIR second derivative spectra minima are far more expressed when compared to their high-temperature counterparts, while the turn/loop motif minima are less expressed, suggesting a slightly different secondary structure.

The seven samples, which showed significant variations in their second derivative FTIR spectra, were further examined by atomic force microscopy. The first notable factor was the difference in fibril self-association (Figure 5A–G, Appendix B, Figure A2A–G and Figure A3A–G). The 25 °C Cluster II, 25 °C Outlier I, 25 °C Outlier II, 45 °C Cluster II and 60 °C Cluster I samples contained aggregates that were highly prone towards binding to one another and forming large fibril clumps, while all other samples were more disperse. Despite the vigorous agitation during aggregation, the 25 °C Outlier II, 40 °C Outlier IV and 60 °C Cluster I samples had relatively long fibrils, suggesting a higher structural stability. Due to certain samples forming large superstructural clusters, conducting a statistical analysis was only possible on their height. A one-way ANOVA Bonferroni means comparison of the height data (*n* = 50, significance level–0.01) revealed that only the 60 °C Cluster I sample had a significantly different height distribution from all other samples, except for the 25 °C Outlier II sample (Figure 5H).

## 3. Discussion

Considering that a shift in sample secondary structure and the existence of high intensity samples coincide with the temperature, at which the prion protein switches between the folded and unfolded states, suggests that this factor is important in determining the type of fibril conformation. Since the temperatures under which the protein is in its mostly folded state also result in significantly slower aggregation, this provides two possibilities. First, certain nuclei may require a relatively long time to form and such stable aggregation centers simply do not have the required time to assemble during quick fibrillization at higher temperatures. Alternatively, it is also possible that distinct aggregates require a specific semifolded state of the protein to both form and grow, which makes their nucleation/elongation events impossible when most of the protein is in a fully unfolded state. In addition, the tested prion protein (89–230 sequence with His-tag) had 54 polar amino acids (33% of the full sequence), 32 of which were charged, which may contribute to the appearance of distinct hydrogen bonds from polar–polar interactions under certain temperature conditions [47].

While such a temperature-dependent shift in fibril conformations can be expected, the truly peculiar aspect is the variation in aggregate secondary structure and morphology. Not a single sample set in the entire tested temperature range contained a well-defined and homogenous collection of fibrils. Each condition resulted in a variable bound-ThT fluorescence intensity, which was well beyond a standard deviation, different EEM maxima positions, FTIR spectra and even fibril morphologies. The large variation of all these factors suggests that prion proteins can form different conformation fibrils or non-homogenous aggregate mixtures under all tested temperatures and this event may encompass other conditions as well. In essence, this means that the assumption of identical conditions leading to identical samples is not correct for prion proteins and this factor has to be taken into account during assays which involve these proteins.

Another interesting observation is the existence of outliers in every set of samples. These outliers are few in number, suggesting that their aggregation pathways are either very complex or unlikely under the given conditions, however, they possess significantly different characteristics. Some of them are so distinct from other fibrils that they can be identified even with a simple visual inspection. While all other samples resulted in an opaque pellet after centrifugation, one type of outlier fibrils had a transparent, gel-like appearance and had a higher tendency of entrapping the buffer solution within itself. Other outliers seem to possess higher structural stabilities, as seen from AFM images. The differences in ThT binding characteristics also suggest a different surface morphology or charge, especially in the case of high fluorescence intensity samples.

Taking everything into consideration, it appears that the environmental conditions during prion protein aggregation do not have a strict control over the type of fibril that forms, but rather determine the dominant aggregate conformation with a certain level of variability. None of the tested conditions resulted in homogenous samples and this factor has to be taken into account, especially when conducting screenings for antiamyloid compounds, as different fibril types may have distinct responses to certain drug molecules.

## 4. Materials and Methods

### 4.1. Prion Protein Aggregation

Mouse recombinant prion protein (89–230) was purified as described previously [43], without the His-tag cleavage step, dialyzed in 10 mM sodium acetate (pH 4.0) for 24 h, concentrated to 3.0 mg/mL and stored at −80 °C prior to use. The protein solution was mixed with 50 mM sodium phosphate (pH 6.0) buffers, containing 0 M or 6 M guanidine hydrochloride (GuHCl) and a ThT stock solution (10 mM) to a final reaction solution, containing 2 M GuHCl, 100 µM ThT and 0.5 mg/mL protein concentration. The reaction solutions were distributed to 96-well half-area non-binding plates (cat. No 3881, Fisher Scientific, USA, Hampton, NH, USA) (100 µL final volume, each well contained a 3 mm bead), which were sealed with a Nunc sealing tape. The samples were incubated at a set temperature (range from 25 to 65 °C), with constant 600 RPM agitation (vigorous, fragmentation-inducing agitation was used to achieve complete fibrillization in a reasonable timeframe). Sample fluorescence was scanned every 5 min, using an excitation wavelength of 440 nm and an emission wavelength of 480 nm. Due to the stochastic nature of aggregation, a small number of samples displayed unusual aggregation kinetics (no detectable signal or signal jumps), which made it impossible to determine their kinetic parameters. Due to this reason, 6 samples were removed from every 96-well plate.

The aggregation lag time (t_lag_) was calculated by applying a sigmoidal Boltzmann equation fit to the data. An example is provided as Appendix B, Figure A4. All aggregation kinetic data is available as Appendix A.

### 4.2. ThT-Assay

After aggregation had occurred, each sample was taken out of the 96-well plate and each well was additionally washed with the reaction solution (50 mM sodium phosphate buffer (pH 6.0) with 2 M GuHCl and 100 µM ThT) in order to collect all the fibrils. The samples were then further diluted using the reaction buffer solution to a final volume of 500 µL (5-fold dilution). Samples were then sonicated for 30 s using a Bandelin Sonopuls (Berlin, Germany) Ultrasonic homogenizer, equipped with a MS-72 tip (20% power). Each sample‘s excitation–emission matrix (EEM) was then scanned using a Varian Cary Eclipse (Agilent Technologies, Santa Clara, CA, USA) fluorescence spectrophotometer (excitation wavelength range was from 435 to 465 nm, emission wavelength range—from 460 to 500 nm; both excitation and emission slit widths were 5 nm) at 25 °C. Absorbance spectra were acquired from 300 to 600 nm using a Shimadzu (Kyoto, Japan) UV-1800 spectrophotometer. EEMs were then corrected for the inner filter effect and their signal intensity centre of mass was calculated as described previously [48]. All EEM data is available as Appendix A. Sample maximum fluorescence intensity distributions were compared using a one-way ANOVA Bonferroni means comparison with a significance level of 0.01 (Origin 2018 software, OriginLab Corporation, Northampton, MA, USA).

### 4.3. Fourier-Transform Infrared Spectroscopy

In order to obtain a larger quantity of fibrils for higher quality FTIR spectra, each sample was combined with a non-aggregated protein solution (identical to the one used for initial aggregation) in equal volumes. Final reaction solutions contained 0.25 mg/mL prion protein and 0.05 mg/mL fibrils (assuming a 100% fibrillization and complete aggregate recovery from wells). The lower non-aggregated protein concentration was used to minimize nucleation events and the relatively high concentration of fibrils assured quick self-replication [24]. The reaction solutions were then incubated under conditions, which were identical to their respective aggregate preparation conditions, in order to achieve effective self-replication.

The resulting aggregate solutions were centrifuged for 30 min at 10,000× *g*, after which the supernatant was removed and the fibril pellet was resuspended into 250 µL D_2_O (D_2_O was supplemented with 100 mM NaCl, which improves fibril sedimentation [49]). This centrifugation and resuspension procedure was repeated 4 times. After the final centrifugation step, the pellet was resuspended into 50 µL D_2_O. The FTIR spectra were acquired as described previously [28]. A D_2_O spectrum was subtracted from each sample spectrum, after which they were baseline corrected and normalized between 1595 and 1700 cm^−1^. All data processing was done using GRAMS software and all FTIR data is available as Appendix A.

### 4.4. Atomic Force Microscopy

Fibril samples were agitated by mixing for 30 s in order to reduce aggregate clumping. Afterwards, 20 µL of each sample was deposited on freshly-cleaved mica and left to adsorb for 60 s. Then the mica were rinsed with 2 mL of H_2_O and air-dried. AFM images were then acquired as previously described [28]. In short, 1024 pixel × 1024 pixel resolution three-dimensional maps were obtained for each sample using a dimension icon (Bruker, Billerica, MA, USA) atomic force microscope. The images were then flattened using Gwyddion 2.5.5 software. Fibril height was determined from line profiles taken perpendicular to the fibril axes. Fibril height distribution was compared using a one-way ANOVA Bonferroni means comparison with a significance level of 0.01 (Origin 2018 software, OriginLab Corporation, Northampton, MA, USA).

## Figures and Tables

**Figure 1 ijms-22-05075-f001:**
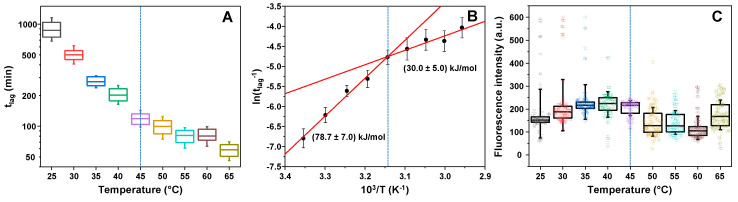
Prion protein aggregation t_lag_ and fibril-bound thioflavin-T (ThT) fluorescence intensity dependence on the aggregation reaction temperature. t_lag_ dependence on the reaction temperature (**A**) and the same data displayed in an Arrhenius plot (**B**), with reaction activation energies shown below linear fit curves. Sonicated and diluted fibril-ThT sample fluorescence intensity distribution (at 25 °C) dependence on the reaction temperature (**C**). Each sample set contained 90 data points, obtained by tracking the aggregation and fluorescence intensity of 90 prion protein aggregation reactions simultaneously at a set temperature. In A and C graphs, box plots indicate the interquartile range and error bars are for one standard deviation. Graph B points are the average ln(t_lag_^−1^) values calculated from the t_lag_ values and error bars are one standard deviation. Dashed blue lines correspond to the temperature at which a non-linear change in t_lag_ and ThT fluorescence intensity is observed. Red lines (**B**) are linear fits of 25–45 °C and 45–65 °C temperature data points. Aggregation t_lag_ box plots (**A**) and their respective sample fluorescence intensities (**C**) are colour-coded.

**Figure 2 ijms-22-05075-f002:**
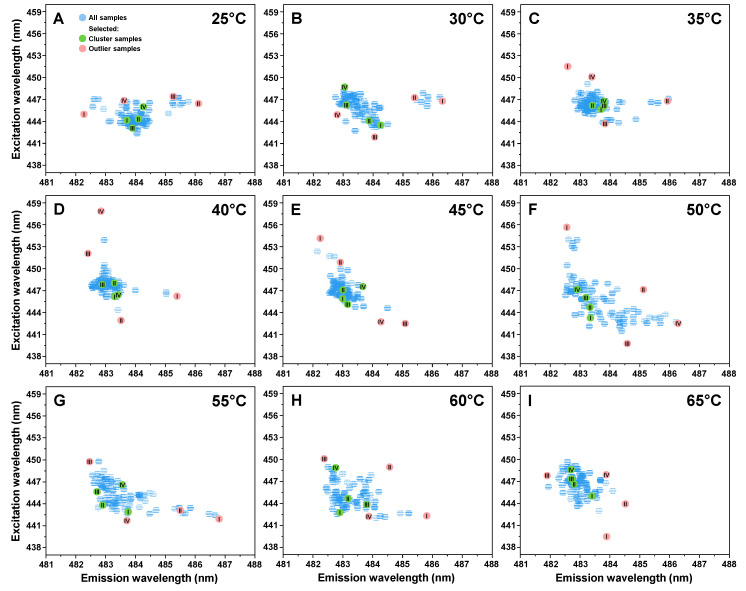
Prion protein fibril-bound thioflavin-T (ThT) fluorescence excitation-emission matrix “center of mass“ distribution. Excitation-emission matrix (EEM) “centers of mass“ were calculated for samples prepared under 25 °C (**A**), 30 °C (**B**), 35 °C (**C**), 40 °C (**D**), 45 °C (**E**), 50 °C (**F**), 55 °C (**G**), 60 °C (**H**) and 65 °C (**I**) temperature conditions as described in the Materials and Methods section. All samples are marked in blue, while samples selected for further analysis from the main cluster and outliers are marked green and red with Roman numerals respectively. Each EEM set was obtained by scanning 90 individual prion protein fibril samples after they were diluted, sonicated and set to 25 °C.

**Figure 3 ijms-22-05075-f003:**
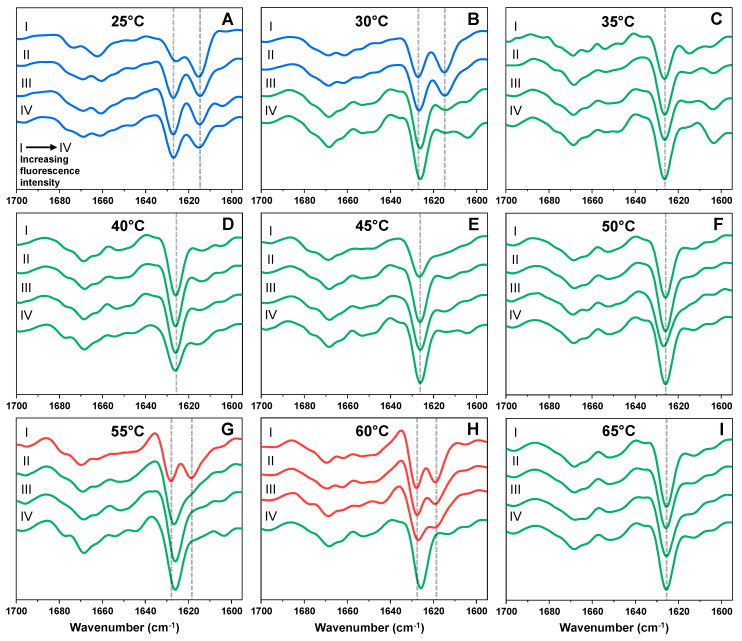
Second derivatives of Fourier-transform infrared spectra (FTIR) of prion protein fibril samples from their respective excitation-emission matrix (EEM) cluster. FTIR spectra were acquired for fibrils prepared under 25 °C (**A**), 30 °C (**B**), 35 °C (**C**), 40 °C (**D**), 45 °C (**E**), 50 °C (**F**), 55 °C (**G**), 60 °C (**H**) and 65 °C (**I**) temperature conditions. The roman numerals indicate the sample ThT-fluorescence intensity, with I being the sample with the lowest emission intensity and IV–the largest. Spectra are colour-coded blue, green and red based on their main minima position similarity. Dotted grey lines indicate the main spectra positions, where differences are observed.

**Figure 4 ijms-22-05075-f004:**
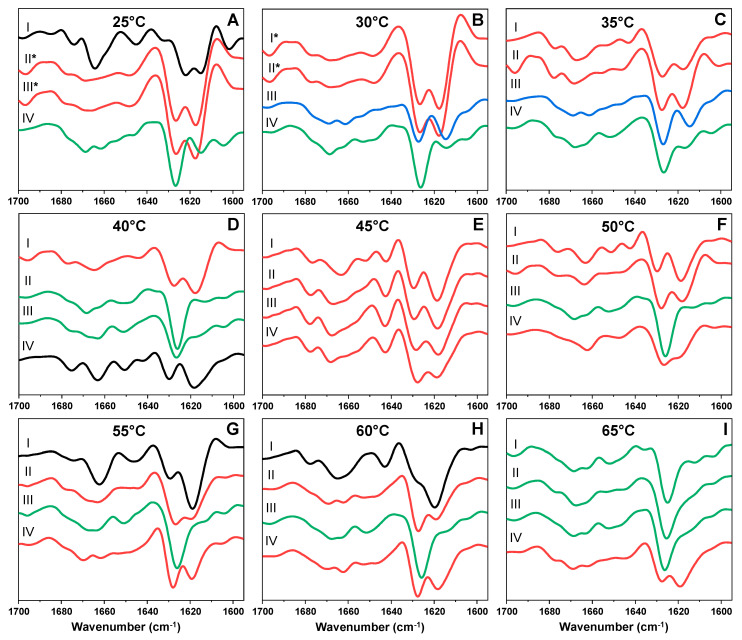
Second derivatives of Fourier-transform infrared spectra (FTIR) of prion protein fibril samples, which were outliers Figure. 25 °C (**A**), 30 °C (**B**), 35 °C (**C**), 40 °C (**D**), 45 °C (**E**), 50 °C (**F**), 55 °C (**G**), 60 °C (**H**) and 65 °C (**I**) temperature conditions. The Roman numerals indicate the outlier sample (list of outlier sample excitation-emission matrix (EEM) positions is located at the Appendix B Table A1). Spectra that share similarities to the main cluster samples are colour-coded accordingly (blue, green and red), while black spectra are unique outliers. The second derivative FTIR spectra of these gel-like samples are marked with a *.

**Figure 5 ijms-22-05075-f005:**
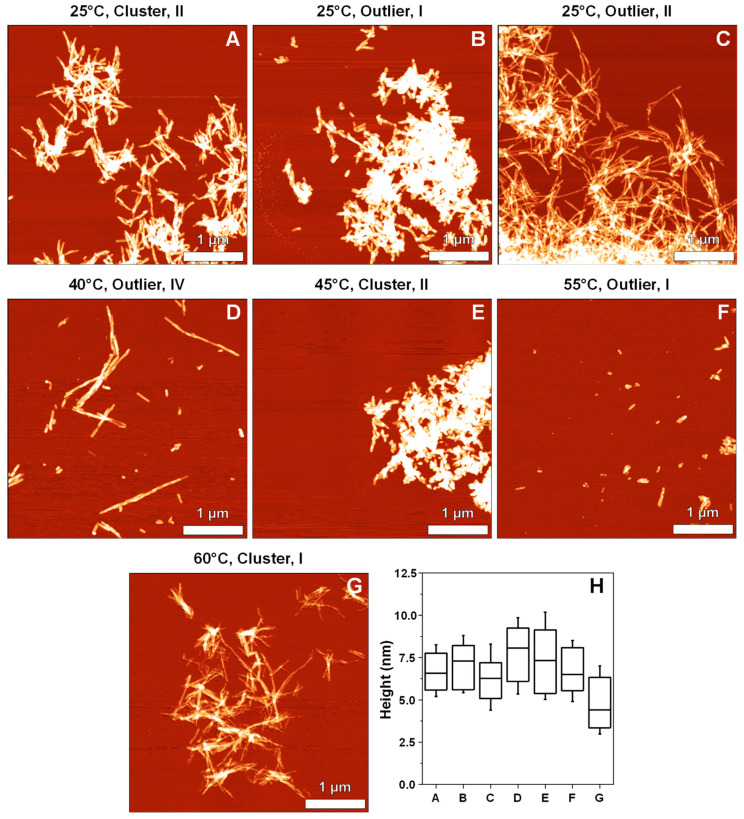
Atomic force microscopy images of prion protein fibrils possessing distinct Fourier-transform infrared (FTIR) spectra ((**A**–**G**), conditions and sample type shown above images). Fibril height (**H**) distribution, where box plots indicate the interquartile range and error bars are one standard deviation (*n* = 50). Fibril height values were obtained by tracing perpendicular to each fibril’s axis (only separate, non-clumped aggregates were examined).

## Data Availability

The data presented in this study are available in Appendix A.

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
