# Peer review of "Temperature-Dependent Structural Variability of Prion Protein Amyloid Fibrils"

_ijms, 2021, doi:10.3390/ijms22105075_

Round 1
Reviewer 1 Report
The article by Ziaunys et al. discusses the structural characterization of prion aggregation in a range of temperatures aiming to describe large aggregate/fibrils. Before supporting for publication I have some comments which can help author to improve the manuscript:
1) Line 37: The concept of protein stability is very broad and indeed it can be associated with a given protein conformation. I suggest to expand the introduction section. In order to answer: what kind of stability is associated with fibrils? generally, one can find thermostability and mechanical stability playing a major role. It is well-known that several biological fibrils including beta-amyloids and alpha-synuclein gain mechanical stability from the oligomer state toward the fibril-like structures (see https://doi.org/10.1039/C7CP05269C and https://doi.org/10.3762/bjnano.10.51) and veryfied by experiment ( Ruggeri et al. https://doi.org/10.1002/ange.201409050)
2) Line 49-51: In recent years, molecular dynamics (MD) simulation has been become a new tool in molecular science. As it can reveal new molecular features of biological fibrils. For instance, the relationship between the fbril symmetry and its stiffness (https://doi.org/10.3762/bjnano.10.51)
3) Line 146: The stabilising polar-polar interactions can give rise to certain hydrogen bonds (HB), as it was identified in the study. If the sequence of the prion is known one can report briefly the number of polar amino acids (aa) and charged aa present in the sequence.
4) In general, it is optional, but one may need an intro cartoon (style TOC) in the first section, so the topic can be introduced.
Author Response
please find response notes in attached file

Reviewer 2 Report
Dear Editor,
The manuscript by Ziaunys et al reports the formation of more than one fibril type over a wide range of temperatures with variability may depend on the state of the initial prion protein molecules.
The design of the study and the technical quality of the work look somehow convincing and results can be of general interest. However, there is a number of major and minor points that would need to be addressed in order to improve the quality of this paper before it can be accepted for publication:
Major:
-The manuscript lacks a clear description about the use of statistical analysis and how it has been performed. Moreover, authors need to perform a post hoc correction to moderate the variability in comparing multiple samples. Bonferroni correction or Conover-Inman post hoc corrections must be performed. Authors need to provide the new corrected p values and report any difference in their observations.
-Authors didn’t provide a reasoning behind their selection for the temperature range especially since the range is outside the physiological limits. While this could be interesting from the biophysical aspect but it has little to do with the pathophysiology underlying the Prion disease. Please elaborate.
Minor:
-Lines 52-54 “The methods used for this, namely atomic force microscopy (AFM) [32] and Fourier- 52 transform infrared spectroscopy (FTIR) [34], are difficult to apply in situations, where a large number of samples have to be differentiated”. Authors have mentioned some of the methods which are currently used. However, neurodegenerative diseases (NDs) are yet incurable conditions. In the discussion, they need to point to out to some of the recent advances in applied for NDs such as the use of high-throughput screening and computer-aided drug design as have been nicely reviewed by Aldewachi et al 2021 and Salman et al 2021 as they can provide a novel insight that can support the findings from this work in future studies. References to be included:
https://pubmed.ncbi.nlm.nih.gov/33672148/
https://www.mdpi.com/1422-0067/22/9/4688
-Figure legends (such as Fig1) lack the mention of actual n number. Authors need to include this information and specify the type of their replicates whether they are biological or statistical.
Best.
Author Response

(The authors gave the same response as above.)

Reviewer 3 Report
In this works the authors show that mouse prion protein can form amyloid fibrils of different properties and this variability is not the result of the difference in experimental conditions. The authors run mouse prion protein aggregation in 96-well plate in the presence of widely used amyloid-intercalating dye ThT and compared maxima in fluorescence emission and excitation spectra for each individual sample. This experiment was repeated at 9 different temperatures ranging from 25 to 65C. About 10% of samples show statistically significant ~2nm red shift in the ThT emission maxima. Fibrils from such outlier samples were compared with the most abundant fibril type using FTIR and AFM. FTIR shows clear difference in the fibril properties and prove the authors` conclusion about variability of fibril type as an intrinsic property of the system.
The overall quality of the experiments and the interpretation is high. The findings are quite interesting but not really novel for the field of amyloid proteins. I think, the authors found an interesting phenomenon that deserves a publication and some additional studies.
Comments
- The aggregation experiments were performed under conditions that induce strong fibril fragmentation (continuous shaking, beads in wells). This masks many morphological properties of fibrils: average fibril length, persistent length, presence and periodicity of fibril height variation. As the result, AFM data is not really informative.
- It would be nice perform an additional aggregation experiment without glass beads in the wells and show the resulted ThT kinetic curves and their analysis in the paper. Fig 3c show that at low temperatures there is a minor population of samples with 2-3x stronger fluorescence intensity. If this high fluorescence intensity also correlates with the ThT maxima position and the average time of fibril duplication during the exponential growth phase, it would be a strong support of the stated fibril polymorphism.
- Fig 2b show two clear subpopulations ("486nm" and "483-484nm" groups). However, only 2 out of 4 samples selected as "outliers" belong to "486nm" group. Do IR spectra of these 2 samples differ from the two other outliers selected from 30C experiment?
- It is quite difficult to see if there is any correlation between the EEM and FTIR data. In Fig 3 curves are labeled according to ThT intensities while Fig 2(that shows selected samples) does not these data. Meanwhile, Fig 1c with ThT intensity data does not show which samples were selected for FTIR studies.
- "Gel-like samples" with much higher viscosity can be formed by longer fibrils that form more rigid network. In this respect it would be logical to discuss the fibril length more. The experiment without glass beads should yield longer fibrils and, therefore, increase the fraction of gel-like samples if the assumption of the fibril length role is correct.
- AFM is not the best method to determine amyloid fibril with, especially for samples like one in Fig 5e. Therefore, I think, Fig 5i and its discussion are misleading.
General comment
Extensive discussion of numerous individual samples with names like "25C Cluster II" scatters the attention. I think the paper would be easier to read and more straightforward if the authors would discuss less temperature points and do a deeper the analysis of samples obtained at one particular temperature instead. (Please treat this comment as an advice for the future works and not as a request to reorganize this manuscript).
Author Response

(The authors gave the same response as above.)

Round 2
Reviewer 2 Report
Dear Editor,
The authors have successfully addressed the majority of my previous comments in order to improve the quality of the manuscript.
I do believe that the corrections, additional sections and updated references, have contributed to enhancing the clarity of the manuscript, which I can endorse for publication.
All the best!
Author Response
-
Reviewer 3 Report
The authors performed the requested experiment and implemented the changes to the manuscript that answers most of my criticism.
I still disagree with some approaches in the experiment planning, but think it is not the reason to delay the publication.
Meanwhile, I would ask the authors to state clearly in the manuscript that the experiment was performed under conditions promoting strong fibril fragmentation and shortly explain the reason.
Author Response
We have added a statement that vigorous, fragmentation-inducing agitation was used to achieve complete fibrillization in a reasonable time frame in both the Introduction and Method sections.